ADID-UNET—a segmentation model for COVID-19 infection from lung CT scans

http://orcid.org/0000-0003-1505-3159 Joseph Raj Alex Noel 1
Zhu Haipeng 1
http://orcid.org/0000-0003-3620-3048 Khan Asiya 2
Zhuang Zhemin 1 zmzhuang@stu.edu.cn
Yang Zengbiao 1
Mahesh Vijayalakshmi G. V. 3
Karthik Ganesan 4
1 Department of Electronic Engineering, College of Engineering, Shantou University , Shantou , China
2 School of Engineering, Computing and Mathematics, University of Plymouth , Plymouth , UK
3 Department of Electronics and Communication, BMS Institute of Technology and Management , Bangalore , India
4 COVID CARE - Institute of Orthopedics and Traumatology, Madras Medical College , Chennai , India
Khan Faizal
Electronic publication date: 2021 Jan 26
Publication date: 2021
Volume: 7
Electronic Location ID: e349
Received 2020 Sep 10; Accepted 2020 Dec 7
Copyright: © 2021 Joseph Raj et al.
Copyright year: 2021
Copyright holder: Joseph Raj et al.
License: This is an open access article distributed under the terms of the Creative Commons Attribution License, which permits unrestricted use, distribution, reproduction and adaptation in any medium and for any purpose provided that it is properly attributed. For attribution, the original author(s), title, publication source (PeerJ Computer Science) and either DOI or URL of the article must be cited.
License URL: https://creativecommons.org/licenses/by/4.0/

Keywords: COVID-19 pulmonary infection, Dense network, Attention gate, Improved dilation convolution, UNET, Lung CT segmentation

Funding: Shantou University, China NTF17016 The National Natural Science Foundation of China 82071992 Guangdong Province Science and Technology 2015B020233018 Guangdong Province University Priority Field (Artificial Intelligence) Project 2019KZDZX1013 This work is supported by The Scientific Research Grant of Shantou University, China (No: NTF17016), The National Natural Science Foundation of China (No. 82071992), The Key Project of Guangdong Province Science and Technology Plan (No. 2015B020233018) and Guangdong Province University Priority Field (Artificial Intelligence) Project (No. 2019KZDZX1013). The funders had no role in study design, data collection and analysis, decision to publish, or preparation of the manuscript.

==============================
Currently, the new coronavirus disease (COVID-19) is one of the biggest health crises threatening the world. Automatic detection from computed tomography (CT) scans is a classic method to detect lung infection, but it faces problems such as high variations in intensity, indistinct edges near lung infected region and noise due to data acquisition process. Therefore, this article proposes a new COVID-19 pulmonary infection segmentation depth network referred as the Attention Gate-Dense Network- Improved Dilation Convolution-UNET (ADID-UNET). The dense network replaces convolution and maximum pooling function to enhance feature propagation and solves gradient disappearance problem. An improved dilation convolution is used to increase the receptive field of the encoder output to further obtain more edge features from the small infected regions. The integration of attention gate into the model suppresses the background and improves prediction accuracy. The experimental results show that the ADID-UNET model can accurately segment COVID-19 lung infected areas, with performance measures greater than 80% for metrics like Accuracy, Specificity and Dice Coefficient (DC). Further when compared to other state-of-the-art architectures, the proposed model showed excellent segmentation effects with a high DC and F1 score of 0.8031 and 0.82 respectively.

Introduction

COVID-19 has caused a worldwide health crisis. The World Health Organization (WHO) announced COVID-19 as a pandemic on March 11, 2020. The clinical manifestations of COVID-19 range from influenza-like symptoms to respiratory failure (i.e., diffuse alveolar injury) and its treatment requires advanced respiratory assistance and artificial ventilation. According to the global case statistics from the Center for Systems Science and Engineering (CSSE) of Johns Hopkins University (JHU) (Wang et al., 2020a) (updated August 30, 2020), 24,824,247 confirmed COVID-19 cases, including 836,615 deaths, have been reported so far with pronounced effect in more than 180 countries. COVID-19 can be detected and screened by Reverse Transcription Polymerase Chain Reaction (RT-PCR). However, the shortage of equipment and the strict requirements on the detection environment limit the rapid and accurate screening of suspected cases. Moreover, the sensitivity of RT-PCR is not high enough, resulting in a large number of false-negatives (Ai et al., 2020), which presents early detection and treatment of patients with presumed COVID-19 (Fang et al., 2020). As an important supplement to RT-PCR, CT scans clearly describe the characteristic lung manifestations related to COVID-19 (Chung et al., 2020), the early Ground Glass Opacity (GGO), and late lung consolidation are shown in Fig. 1. Nevertheless, CT scans also show imaging features that are similar to other types of pneumonia, making it difficult to differentiate them. Moreover, the manual depiction of lung infection is a tedious and time-consuming job, which is often influenced by personal bias and clinical experience.

Figure 1 (A) and (C) represent the CT images and (B) and (D) correspond to the COVID-19 infected areas in CT axial section.

Here the blue and yellow masks represent the Ground Glass Opacity(GGO) and the late lung consolidation segments respectively. The images were obtained from (MedSeg, 2020).

In recent years, deep learning has been gaining popularity in the field of medical imaging due to it’s intelligent and efficient feature extraction ability (Kong et al., 2019; Ye, Gao & Yin, 2019), and has achieved great success. An earliest classic example is the application of deep learning to children’s chest X-rays to detect and distinguish bacterial and viral pneumonia (Kermany et al., 2018; Rajaraman et al., 2018). Also using deep learning methods have been applied to detect various imaging features of chest CT images (Depeursinge et al., 2015; Anthimopoulos et al., 2016). Recently, researchers proposed to detect COVID-19 infections in patients by radiation imaging combined with deep learning technology. Li et al. (2020) proposed a simple Cov-Net deep learning network in combination with a deep learning algorithm, which was used to distinguish COVID-19 and Community-Acquired Pneumonia (CAP) from chest CT scans. Wang & Wong (2020) proposed Covid-Net to detect COVID-19 cases from chest X-ray images, with an accuracy rate of 93.3%. The infection probability of COVID-19 Xu et al. (2020) was calculated from CT scans by adopting a position-oriented attention model that presented accuracy close to 87%. However, the above models rarely involved the segmentation of COVID-19 infection (Chaganti et al., 2020; Shan et al., 2020). The challenges involved in segmentation include: (a) variations in texture, size, and position of the infected areas in CT scans. For example, some infection areas are small, which easily lead to a high probability of false negatives in CT scans. (b) The boundary of GGO is usually of low contrast and fuzzy in appearance, which makes it difficult to distinguish from the healthy regions during the segmentation process. (c) The noise around the infected area is high, which greatly affects the segmentation accuracy and (d) finally the cost and time consumed in obtaining high-quality pixel-level annotation of lung infection in CT scans is high. Therefore, most of the COVID-19 CT scan datasets are focused on diagnosis, and only a few of them provide segmentation labels. However, with the passage of time, the annotated datasets for the segmentation of COVID-19 pulmonary infection were released but due to a lesser amount of data, the phenomenon of overfitting could cause problems while training thus necessitating the need for more segmentation datasets and better algorithms for accurate results.

Therefore, to address the challenges stated above, we propose a new deep learning network called Attention Gate-Dense Network- Improved Dilation Convolution-UNET (ADID-UNET) for the segmentation of COVID-19 from lung infection CT scans. Experimental results on a publicly available dataset illustrate that the proposed model presents reliable segmentation results that are comparable to the ground truths annotated by experts. Also, in terms of performance, the proposed model surpasses other state-of-the-art segmentation models, both qualitatively and quantitatively.

Our contributions in this paper are as follows:To address the problem that the gradient disappearance in the deep learning network pose, we employ a dense network (Huang et al., 2017) instead of a traditional convolution and max-pooling operations. The dense network extracts dense features and enhances feature propagation through the model. Moreover, the training parameters of the dense network are less, which reduces the size and the computational cost.

To increase the size of the respective field and to compensate for the problems due to blurry edges, an improved dilation convolution (IDC) module is used to connect the encoder and decoder pipelines. The IDC model increases the receptive field of the predicted region providing more edge information, which enhances the edge recognition ability of the model.

Since the edge contrast of GGO is very low, we use the attention gate (AG) instead of simple cropping and copying. This further improves the accuracy of the model to detect the infection areas by learning the characteristics of the infected regions.

Due to the limited number of COVID-19 segmented datasets with segmentation labels, which is less than the minimum number of samples required for training a complex model, we employ data augmentation techniques and expand the dataset on the basis of the collected public datasets.

The rest of the paper is organized as follows: “Related Work” describes the work related to the proposed model. “Methods” introduces the basic structure of ADID-UNET. Details of the dataset, experimental results and discussion are dealt with in “Experimence Results”. Finally, “Conclusion” presents the conclusion.

Related work

ADID-UNET model proposed in this paper is based on UNET (Ronneberger, Fischer & Brox, 2015) architecture and therefore, we will discuss the literature related to our work which includes: deep learning and medical image segmentation, improvement of medical image segmentation algorithms, CT scan segmentation, and application of deep learning in segmentation of COVID-19 lesions from lung CT scans.

Deep learning and medical image segmentation

In recent years, deep learning algorithms have become more mature leading to various artificial intelligence (AI) systems based on deep learning algorithms being developed. Also, semantic segmentation using deep learning algorithms (Oktay et al., 2018) has developed rapidly with applications in both natural and medical images. Long, Shelhamer & Darrell (2015) pioneered the use of a fully connected CNN (FCN) to present rough segmentation outputs that were of the input resolution through fractionally strided convolution process also referred as the upsampling or deconvolution. The model was tested on PASCAL VOC, NYUDv2, and SIFT datasets and, presented a Mean Intersection of Union (M-IOU) of 62.7%, 34%, 39.5%, respectively. They also reported that upsampling, part of the in-network, was fast, accurate, and provided dense segmentation predictions. Later through a series of improvements and extensions to FCN (Ronneberger, Fischer & Brox, 2015; Badrinarayanan, Kendall & Cipolla, 2017; Xu et al., 2018), a symmetrical structure composed of encoder and decoder pipelines, called UNET (Ronneberger, Fischer & Brox, 2015), was proposed for biomedical or medical image segmentation. The encoder structure predicted the segmentation area, and then the decoder recovered the resolution and achieved accurate spatial positioning. Also, the UNET used crop and copy operations for the precise segmentation of the lesions. Further, the model achieved good segmentation performance at the International Symposium on Biomedical Imaging (ISBI) challenge (Cardona et al., 2010) with the M-IOU of 0.9203. Moreover, an improved network referred as the SegNet was proposed by Badrinarayanan, Kendall & Cipolla (2017). The model used the first 13 convolution layers of the VGG16 network (Karen & Andrew, 2014) to form an encoder to extract features and predict segmentation regions. Later by using a combination of convolution layers, unpooling and softmax activation function in the decoder, segmentation outputs of input resolution were obtained. When tested with the CamVid dataset (Brostow, Fauqueur & Cipolla, 2009), the M-IOU index of SegNet was nearly 10% higher than that of FCN (Long, Shelhamer & Darrell, 2015). Xu et al. (2018) regarded segmentation as a classification problem in which each pixel was associated with a class label and designed a CNN network composed of three layers of convolution and pooling, a fully connected layer (FC) and softmax function. The model of successfully segmented three-dimensional breast ultrasound (BUS) image datasets was presented into four parts: skin, fibroglandular tissue, mass, and fatty tissue and achieved a recall rate of 88.9%, an accuracy of 90.1%, precision of 80.3% and F1 score of 0.844. According to the aforementioned literature, FCN (Long, Shelhamer & Darrell, 2015) and their improved variants presented accurate segmentation results for both natural or medical images. Therefore, the UNET and variants (Almajalid et al., 2019; Negi et al., 2020), due to its advantages of fast training and high segmentation accuracy are widely used in the field of medical image segmentation.

Improvement of medical image segmentation algorithms

Medical images such as the ultrasound images are generally prone to speckle noise, uneven intensity distribution, and low contrast between the lesions and the backgrounds which affect the segmentation ability of the traditional UNET (Ronneberger, Fischer & Brox, 2015) structure. Therefore, considerable efforts were invested in improving the architecture. Xia & Kulis (2017) proposed a fully unsupervised deep learning network called W-Net model that connects two UNETs to predict and reconstruct the segmentation results. Schlemper et al. (2019) proposed an attention UNET network, which integrated attention modules into the UNET (Ronneberger, Fischer & Brox, 2015) model to achieve spatial positioning and subsequent segmentation. The model presented a segmentation accuracy of 15% higher than the traditional UNET architecture. Zhuang et al. (2019a) combined the goodness of the attention gate system and the dilation convolution module and proposed a hybrid architecture referred as the RDA-UNET. By introducing residual network (He et al., 2016) instead of traditional convolution layers they reported a segmentation accuracy of 97.91% towards the extraction of lesions in breast ultrasound images. Also, the GRA-UNET (Zhuang et al., 2019b) model included a group convolution module in-between the encoder and decoder pipelines to improve the segmentation of the nipple region in breast ultrasound images. Therefore, from the literature, it can be inferred that introducing additional modules like attention gate instead of traditional cropping and copying, inclusion of dilation convolution to increase the receptive fields and use of residual networks can favorably improve the accuracy of the segmentation model. However, these successful segmentation models (Schlemper et al., 2019; Zhuang et al., 2019a; Xia & Kulis, 2017) were rarely tested with CT scans, hence the next section concentrates on the segmentation of CT scans.

CT scan segmentation

CT imaging is a commonly used technology in the diagnosis of lung diseases since lesions can be segmented more intuitively from the chest CT scans. The segmented lesion aid the specialist in the diagnosis and quantification of the lung diseases (Gordaliza et al., 2018). In recent years, most of the classifier models and algorithms based on feature extraction have achieved good segmentation results in chest CT scans. Ye et al. (2009) proposed a shape-based Computer-Aided Detection (CAD) method where a 3D adaptive fuzzy threshold segmentation method combined with chain code was used to estimate infected regions in lung CT scans. In feature-based techniques, due to the low contrast between nodules and backgrounds, the boundary discrimination is unclear leading to inaccurate segmentation results. Therefore, many segmentation techniques based on deep learning algorithms have been proposed. Wang et al. (2017) developed a central focusing convolutional neural network for segmenting pulmonary nodules from heterogeneous CT scans. Jue et al. (2018) designed two deep networks (an incremental and dense multiple resolution residually connected network) to segment lung tumors from CT scans by adding multiple residual flows with different resolutions. Guofeng et al. (2018) proposed a UNET model to segment pulmonary nodules in CT scans which improved the overall segmentation output through the avoidance of overfitting. Compared with other segmentation algorithms such as graph-cut (Ye, Beddoe & Slabaugh, 2009), their model had better segmentation results with a Dice coefficient of 0.73. Recently, Peng et al. (2020) proposed an automatic CT lung boundary segmentation method, called Pixel-based Two-Scan Connected Component Labeling-Convex Hull-Closed Principal Curve method (PSCCL-CH-CPC). The model included the following: (a) the image preprocessing step to extract the coarse lung contour and (b) coarse to finer segmentation algorithm based on the improved principal curve and machine learning model. The model presented good segmentation results with Dice coefficient as high as 96.9%. Agarwal et al. (2020) proposed a weakly supervised lesion segmentation method for CT scans based on an attention-based co-segmentation model (Mukherjee, Lall & Lattupally, 2018). The encoder structure composed of a variety of CNN architectures that includes VGG-16 (Karen & Andrew, 2014), Res-Net101 (He et al., 2016), and an attention gate module between the encoder-decoder pipeline, while decoder composed of upsampling operation. The proposed method first generated the initial lesion areas from the Response Evaluation Criteria in Solid Tumors (RECIST) measurements and then used co-segmentation to learn more discriminative features and refine the initial areas. The paper reported a Dice coefficient of 89.8%. The above literatures suggest that deep learning techniques are effective in segmenting lesions in lung CT scans and many researchers have proposed different deep learning architectures to deal with COVID-19 CT scans. Therefore, in the next section we will further study their related works.

Application of deep learning in segmentation of COVID-19 lesions from lung CT scans

In recent months, COVID-19 has become a hot topic of concern all over the world and CT imaging is considered to be a convincing method to detect COVID-19. However, due to the limited datasets and the time and labor involved in annotations, segmentation datasets related to COVID-19 CT scans are less readily available. But, many researchers have still proposed advanced methods to deal with COVID-19 diagnosis, which also includes segmentation techniques (Fan et al., 2020; Wang et al., 2020b; Yan et al., 2020; Zhou, Canu & Ruan, 2020; Elharrouss et al., 2020; Chen, Yao & Zhang, 2020). On the premise of insufficient datasets with segmentation labels, the Inf-Net network proposed by Fan et al. (2020), combined a semi-supervised learning model and FCN8s network (Long, Shelhamer & Darrell, 2015) with implicit reverse attention and explicit edge attention mechanism to improve the recognition rate of infected areas. The model successfully segmented COVID-19 infected areas from CT scans and reported a sensitivity and accuracy of 72.5% and 96.0%, respectively. Elharrouss et al. (2020) proposed an encoder-decoder-based CNN method for COVID-19 lung infection segmentation based on a multi-task deep-learning based method, which overcame the shortage of labeled datasets, and segmented lung infected regions with a high sensitivity of 71.1%. Wang et al. (2020b) proposed a noise-robust COVID-19 pneumonia lesions segmentation network which included a noise-robust dice loss function along with convolution function, residual network, and Atrous Spatial Pyramid Pooling (ASPP) module. The model was referred as Cople-Net presented automatic segmentation of COVID-19 pneumonia lesions from CT scans. The method proved that the proposed new loss function was better than the existing noise-robust loss functions such as Mean absolute error (MAE) loss (Ghosh, Kumar & Sastry, 2017) and Generalized Cross-Entropy (GCE) loss (Zhang & Sabuncu, 2018) and achieved a Dice coefficient and Relative Volume Error (RVE) of 80.72% and 15.96%, respectively. Yan et al. (2020) employed an encoder-decoder deep CNN structure composed of convolution function, Feature Variation (FV) module (mainly contains convolution, pooling, and sigmoid function), Progressive Atrous Spatial Pyramid Pool (PASPP) module (including convolution, dilation convolution, and addition operation) and softmax function. The convolution function obtained features, FV block enhanced the feature representation ability and the PASPP was used between encoder and decoder pipelines compensated for the various morphologies of the infected regions. The model achieved a good segmentation performance with a Dice coefficient of 0.726 and a sensitivity of 0.751 when tested on the COVID-19 lung CT scan datasets. Zhou, Canu & Ruan (2020) proposed an encoder-decoder structure based UNET model for the segmentation of the COVID-19 lung CT scan. The encoder structure was used to extract features and predict rough lesion areas which composed convolution function and Res-dil block (combines residual block (He et al., 2016) and dilation convolution module). The decoder pipeline was used to restore the resolution of the segmented regions through the upsampling and the attention mechanism between the encoder-decoder framework to capture rich contextual relationships for better feature learning. The proposed method can achieve an accurate and rapid segmentation on COVID-19 lung CT scans with a Dice coefficient, sensitivity, and specificity of 69.1%, 81.1%, and 97.2%, respectively. Further, Chen, Yao & Zhang (2020) proposed a residual attention UNET for automated multi-class segmentation of COVID-19 lung CT scans, which used residual blocks to replace traditional convolutions and upsampling functions to learn robust features. Again, a soft attention mechanism was applied to improve the feature learning capability of the model to segment infected regions of COVID-19. The proposed model demonstrates a good performance with a segmentation accuracy of 0.89 for lesions in COVID-19 lung CT scans. Therefore, the deep learning algorithms are helpful in segmenting the infected regions from COVID-19 lung CT scans which aid the clinicians to evaluate the severity of infection (Tang et al., 2020), large-scale screening of COVID-19 cases (Shi et al., 2020) and quantification of the lung infection (Ye et al., 2020). Table 1 summarizes the deep learning-based segmentation techniques available for COVID-19 lung infections.

Table 1 The summary of various deep learning algorithms for COVID-19 lung CT scans and the segmentation results.

RVE, ACC, DC, Sen, Sp and F1 represent relative volume error, accuracy, Dice coefficient, sensitivity, specificity and F1 score, respectively.

Literature	Data Type	Dataset	Technique	Segmentation results	
Fan et al. (2020)	CT Scan	100 CT images	Semi supervised CNN	73.9% (DC)	
			FCN8s network	96.0% (Sp)	
Wang et al. (2020b)	CT Scan	558 CT images	Residual connection	80.7% (DC)	
			CNN	16.0% (RVE)	
Yan et al. (2020)	CT Scan	21,658 CT images	Deep CNN	72.6% (DC)	
				75.1% (Sen)	
Zhou, Canu & Ruan (2020)	CT Scan	100 CT images	Attention mechanism	69.1% (DC)	
			Res-Net, dilation convolution	81.1% (Sen)	
Elharrouss et al. (2020)	CT Scan	100 CT images	Encoder-decoder-based CNN	78.6% (Dice)	
				71.1% (Sen)	
Chen, Yao & Zhang (2020)	CT Scan	110 CT images	Encoder-decoder-based CNN	83.0% (DC)	
				89.0% (ACC)	
Xu et al. (2020)	CT Scan	110 CT images	CNN	86.7% (ACC)	
				83.9% (F1)	
Shuai et al. (2020)	CT Scan	670 CT images	CNN	73.1% (ACC)	
				67.0% (Sp)	

Methods

In this section, we first introduce the proposed ADID-UNET network with detailed discussion on the core network components including dense network, improved dilation convolution, and attention gate system. To present realistic comparisons, experimental results are presented at each subsection to illustrate the performance and superiority of the model after adding core components. Further in “Experimence Results” we have presented a summary of the % improvements achieved when compared to the traditional UNET architecture.

ADID-UNET architecture

ADID-UNET is based on UNET (Ronneberger, Fischer & Brox, 2015) architecture with the following improvements: (a) The dense network proposed by Huang et al. (2017) is used in addition to the convolution modules of encoder and decoder structures, (b) an improved dilation convolution (IDC) is introduced between the frameworks, and (c) the attention gate (AG) system is used instead of the simple cropping and copying operations. The structure of ADID-UNET is shown in Fig. 2. Here fen, fupn, fidc describe the features at the n-th layer of the encoder, decoder, and IDC modules, respectively.

Figure 2 The structure of the ADID-UNET network.

The blue, purple, black and orange arrows represent transfer function, convolution function with 1×1 convolution kernel and sigmoid function, upsampling function, convolution function with 3×3 convolution kernel and RELU function, respectively. The triangle represents the attention gate system. The orange, purple, blue and black dotted boxes represent the dense network, concatenate function, dilation convolution and improved dilation convolution, respectively. The curved black arrows within the orange rectangle indicate the dense block. The C within the circle represents concatenate function. The gray, orange, green, black, purple and blue squares represent convolution function, maximum pooling function, improved void convolution layer, attention gate layer, upsampling layer and transition layer, respectively, and describe the features at n-th layer of the encoder, decoder and IDC module, respectively.

When COVID-19 CT scans are presented to the encoder, the first four layers (each layer has convolutions, rectification, and max pooling functions) extract features (f1–f4) that are passed to dense networks. Here dense networks are used instead of convolution and max-pooling layers to further enhance the features (f5–f6) and in “Dense Network”, we elaborate the need for the dense network and present experimental results to prove its significance. Next, an improved dilation convolution module referred as the IDC model, is used between the encoder–decoder structure to increase the receptive field and gather detailed edge information that assists in extracting the characteristic. The module accepts the feature f6 from the dense networks and after improvement, present fidc them as inputs to the decoder structure. To ensure consistency in the architecture and to avoid losing information, the decoder mirrors the encoder with two dense networks that replace the first two upsampling operations. Further for the better use of the context information between the encoder-decoder pipeline, the AG model is used instead of cropping and copying operations, which aggregates the corresponding layer-wise encoder features with the decoder and presents it to the subsequent upsampling layers. Likewise, the decoder framework presents upsampled features fup1 to fup6 and final feature map (fup6) is presented to the sigmoid activation function to predict and segment the COVID-19 lung infected regions. The following section explains the components of ADID-UNET in detail.

Dense network

It was presumed that with the increase of network layers, the learning ability of the network will gradually improve, but during the training, for deep networks, the gradient information that is helpful for the generalization may disappear or expand excessively. In literature, the problem is referred as vanishing or explosion of the gradient. As the network begins to converge, due to the disappearance of the gradient the network saturates, resulting in a sharp decline in network performance. Therefore, Zhuang et al. (2019a) introduced residual units proposed by He et al. (2016) into UNET structure to avoid performance degradation during training. The residual learning correction scheme to avoid performance degradation is described in (1): (1) y=G(x,{Fi})+x

Here x and y are the input and output vectors of the residual block, Fi is the weight of the corresponding layer. The function G(x,{Fi}) is a residue when added to x, avoids vanishing gradient problems, and enables efficient learning.

From (1) the summation of G(x,{Fi}) and x in Res-Net (He et al., 2016) avoids the vanishing gradient problems but forwarding the gradient information alone to the proceeding layers may hinder the information flow in the network and the recent work by Huang et al. (2016) illustrated that of Res-Nets discard features randomly during training. Moreover, Res-Nets include large number of parameters, which increases the training time. To solve this problem, Huang et al. (2017) proposed a dense network (as shown in Fig. 3), which directly connects all layers, and thus skillfully obtains all features of the previous layer without convolution.

Figure 3 Schematic diagram of the dense network.

Green and purple squares represent the input image and the output image, respectively. The blue rectangles represent convolution function and pooling function. The dashed black box represents the dense block and the red and yellow rectangle module within represent convolution function. The pink trapezoidal clock represents the transition function and N represents the number of Dense Block and Transition layer is N.

The dense network is mainly composed of convolution layers, pooling function, multiple dense blocks, and transition layers. Let us consider a network with L layers, and each layer implements a nonlinear transformation Hi. Let x0 represent the input image, i represents layer i, xi−1 is the output of layer i − 1. Hi can be a composite operation, such as batch normalization (BN), rectified linear function (RELU), pooling, or convolution functions. Generally, the output of traditional network in layer i is as follows: (2) xi=Hi×(xi−1)

For the residual network, only the identity function from the upper layer is added: (3) xi=Hi×(xi−1)+xi−1

For a dense network, the feature mapping x0, x1,…, xi−1 of all layers before layer i is directly connected, which is represented by Eq. (4): (4) xi=Hi×([x0,x1,…,xi−1])

where [x0,x1,…,xi−1] denotes the cascade of characteristic graphs and × represents the multiplication operation. Figure 4 shows the forward connection mechanism of the dense network where the output of layers is connected directly to all previous layers.

Figure 4 It is more intuitive to understand the forward connection mode of a dense network.

The forward connection model of the Dense Network, illustrating that the output xi includes inputs from x0, x1,…,xi−1.

Generally, a dense network is composed of several dense blocks and transition layers. Here we only use two dense blocks and transition layers to form simple dense networks. Using Eq. (5) to express the dense block: (5) γ=α([x0,x1,…,xi−1],βi)

where [x0,x1,…,xi−1] denotes the cascade of characteristic graphs, βi is the weight of the corresponding layer. In the ADID-UNET model proposed in this paper, the feature f4 (refer to Fig. 2) is fed to the transition layer, which is mainly composed of BN, RELU, and average pooling operation. Later the feature is batch standardized and rectified before convolving with a 1 × 1 kernel function. Again, the filtered outputs go through the same operation and are convoluted with 3 × 3 kernel, before concatenating with the input feature f4. The detailed structure of the two dense blocks and transition layers used in the encoder structure is shown in Fig. 5A. Here w, h correspond to the width and height of the input, respectively, and b represents the number of channels. Besides, s represents the step size of the pooling operation, n represents the number of filtering operations performed by each layer. In our model, n takes values 32, 64, 128, 256, and 512. It should be noted that the output of the first dense layer is the aggregated result of 4 convolution operations (4 × n), which is employed to emphasize the features learning by reducing the loss of features. In the decoding structure, to restore the resolution of the predicted segmentation, a traditional upsampling layer of the UNET (Ronneberger, Fischer & Brox, 2015) is used instead of the transition layer. The detailed structure is shown in Fig. 5B.

Figure 5 Dense network for encoder and decoder pipelines.

(A) The dense network of the encoder pipeline. (B) The dense network of the decoder pipeline. Here, w, h, b, s, and n correspond to the width and height of the input, the number of channels, the step size of the pooling operation, and the number of filtering operations performed by each layer, respectively. For layers 6, 5, 4, 3, 2, 1, the values of n are 512, 512, 256, 128, 64, and 32, respectively.

For the proposed network, we use only two dense networks mainly (a) to reduce the computation costs and (b) experiments with different layers of dense networks suggest that the use of two dense networks was sufficient since the segmentation results were accurate and comparable to the ground truth. Figure 6 and Table 2 illustrate the qualitative and quantitative comparisons with different numbers of dense network in the encoder-decoder framework.

Figure 6 Results of adding different numbers of dense networks.

Here (A–C) are the CT scans. For each row, (D) denotes the ground truth, (E–J) illustrate the segment results from UNET, Num1, Num2, Num3, Num4 and Res-Net, respectively. Num1∼Num4 denotes the number of dense networks used in the encoder and decoder pipelines. Res-Net refers to the network where convolution operations are replaced by the residual network (He et al., 2016). UNET (Ronneberger, Fischer & Brox, 2015) denotes the traditional architecture without dense network.

Table 2 Quantitative comparisons with respect to ground truth for different dense layers included the UNET (Ronneberger, Fischer & Brox, 2015) and Res-Net (He et al., 2016) architecture.

ACC, DC, Sen, Sp, Pc, AUC, F1, Sm, Ea and MAE represent accuracy, Dice coefficient, sensitivity, specificity, precision, the area under the curve, F1 score, structural metric, enhancement alignment meter and mean absolute error, respectively.

Number of dense network	ACC	DC	Sen	Sp	Pc	AUC	F1	Sm	Eα	MAE	
Num1	0.9696	0.7971	0.8011	0.9958	0.8290	0.9513	0.8129	0.8411	0.9315	0.0088	
Num2	0.9700	0.8011	0.8096	0.9966	0.8596	0.9492	0.8184	0.8528	0.9394	0.0083	
Num3	0.9686	0.7569	0.7546	0.9957	0.8200	0.9334	0.7806	0.8349	0.9379	0.0104	
Num4	0.9699	0.7869	0.7579	0.9961	0.8485	0.9495	0.8241	0.8341	0.9348	0.0090	
UNET	0.9696	0.7998	0.8052	0.9957	0.8247	0.9347	0.8154	0.8400	0.9390	0.0088	
Res-Net	0.9698	0.8002	0.7978	0.9962	0.8344	0.9504	0.8180	0.8415	0.9352	0.0094	

From the analysis of results in Fig. 6 and Table 2, it is found that the effect of using two dense networks in the model is obvious and can present accurate segments of the infected areas that can be inferred directly from the qualitative and quantitative metrics.

Moreover, with high accuracy and a good Dice coefficient, the choice of two dense networks is the best choice in the encoder decoder pipeline. Also, using two dense networks in place of traditional convolutions or residual networks enable global feature propagation, encourage feature reuse, and also solve the gradient disappearance problems associated with deep networks thereby significantly improving the segmentation outcomes.

Improved dilation convolution

Since the encoder pipeline of the UNET structure is analogous to the traditional CNN architecture, the pooling operations involved at each layer propagate either the maximum or the average characteristics of the extracted features, hence connecting the encoder outputs directly to decoder, thus limiting the segmentation accuracy of the network. The RDA-UNET proposed by Zhuang et al. (2019a) utilized a dilation convolution (DC) module between the encoder-decoder pipeline to increase the receptive field and further learn the boundary information accurately. Also, the DC module is often used in many variant UNETs (Chen et al., 2019; Yu & Koltun, 2015) to improve the receptive field, hence, we use the DC module and introduce additional novelty in the DC module.

Equation (6) describes the DC operation between the input image f(x,y) and the kernel g(i,j).

(6) p(x,y)=α{∑i,jf(x+i×r,y+j×r)×g(i,j)+k}

where α is the RELU function, k is a bias unit (i,j) and (x,y) denote the coordinates of the kernel and those of the input images respectively, and r is the dilation rate that controls the size of receptive fields. The size of the receptive field obtained can be expressed as follows: (7) N=((k_fsize+1)×(r−1)+k_fsize)

where k_fsize is the convolution kernel size, r is the convolution rate of the dilation and N is the size of the receptive field. As shown in Fig. 7.

Figure 7 Schematic diagram of dilation convolution.

(A) It shows the visual field effect of the classical 3 × 3 convolution kernel, covering 3 × 3 field of view each time (purple part of figure (A)); (B) corresponds to 3 × 3 with r = 2. Though the size of the convolution kernel is still 3 × 3, but the receptive field of convolution kernel is increased to 7 × 7 (purple and green parts in figure (B)); (C) corresponds to 3 × 3 with r = 3 and a receptive field of 15 × 15.

Based on our experimental analysis we understand that DC module has a pronounced effect in extracting information for larger objects or lesions and considering that most of the early ground-glass opacity (GGO) or late lung consolidation lesions have smaller areas, we present an improved dilation convolution (IDC) module between the encoder–decoder framework to accurately segment smaller regions.

Figure 8 illustrates the IDC module that consists of several convolution functions with different dilation rates and rectified linear functions (RELU). Our improvements are as follows: (a) combining single strided convolution operations and dilated convolutions with dilation rate such as 2, 4, 8, and 16, respectively. The above combination helps in the extraction of features from both smaller and larger receptive fields thus assisting in the isolation of the small infected COVID-19 regions seen in lung CT scans and (b) referring to the idea of the dense network (Huang et al., 2017), we concatenate the input of the IDC module to its output and use the information of input features to further enhance feature learning. The input of IDC module is the rough segmentation regions obtained by encoder structure. The combination of the original segmentation region features and the accurate features extracted by IDC module not only avoids the loss of useful information, but also provides accurate input for the decoding pipeline, which is conducive to improve the segmentation accuracy of the model. As the inputs advance (left to right in Fig. 8), they get convolved with a 3 × 3 kernel of convolution layers and the dilation rate of IDC is 2, 4, 8, and 16, respectively. From the comparative experiments with the traditional DC model (the dilation rate is the same for both the models), we find that the computational cost and computation time required for the IDC module is less than that of the DC module, as shown in Table 3.

Figure 8 Improved dilation convolution (IDC).

The gray, black and green rectangles represent layer 6 features of the encoder, convolution layer and improved dilation convolution layer, respectively. The blue dotted box indicates the dilation convolution. Blue and orange arrows represent the transfer function and the convolution function with the convolution kernel of 3×3 and the RELU function, respectively.

Table 3 Comparison results of the number of parameters of the UNET model with dense network incorporating either improved dilation convolution module (IDC) or dilation convolution module.

Here DID-UNET refers to the inclusion of dense networks and IDC module and DD-UNET denotes dense networks and traditional dilation convolution added to the UNET structure.

Method	Total parameters	Trainable parameters	Non-trainable parameters	Train time epoch/(s)	Test time (s)	
DD-UNET	56,223,034	56,190,272	32,762	145	8	
DID-UNET	52,162,362	52,132,416	29,946	135	3	

From Fig. 9 and Table 4, it is found that the use of layers with convolution and smaller dilation rates at the end along with others ensures the cumulative extraction of features from both smaller and larger receptive fields thus assisting in the isolation of the small infected COVID-19 regions seen in lung CT scans. Also, the performance scores specifically the Dice coefficient is higher (about 3%) for DID-UNET compared to DD-UNET. In summary, the IDC model connected between the encoder–decoder structure, reduces loss of the original features but additionally expands the field of the segmented areas thereby improving the overall segmentation effect.

Figure 9 Experimental results of the improved dilation convolution and the traditional dilation convolution.

Here (A) is the CT scan, (B) is the ground truth, (C–E) are the segment results from UNET, DD UNET and DID UNET, respectively. Here DID-UNET refers to the inclusion of dense networks and IDC module. DD-UNET denotes dense networks with traditional dilation convolution added to the UNET.

Table 4 The results of comparison indees of the improved dilation convolution experiment and the traditional dilation convolution experiment.

Method	ACC	DC	Sen	Sp	Pc	AUC	F1	Sm	Eα	MAE	
UNET	0.9696	0.7998	0.8052	0.9957	0.8247	0.9347	0.8154	0.8400	0.9390	0.0088	
DD-UNET	0.9697	0.7757	0.7402	0.9971	0.8622	0.9214	0.7923	0.8401	0.9312	0.0094	
DID-UNET	0.9700	0.8023	0.7987	0.9964	0.8425	0.9549	0.8241	0.8447	0.9374	0.0084	

Attention gate

Although the improved dilation convolution improves the feature learning ability of the network, due to the loss of spatial information in the feature mapping at the end of the encoder structure, the network has difficulties in reducing false prediction for (a) small COVID-19 infected regions and (b) areas with blurry edges with poor contrast between the lesion and background. To solve this problem, we introduce the attention gate (AG) model shown in Fig. 10 mechanism into our model instead of simple cropping and copying. AG model computes the attention coefficient σ∈[0,1], based on Eq. (8): (8) σ=ε2{pk[pi(ε1(pn×n+pm×m+bm,n))+bint]+bk}

(9) ε2(x)=11+exp⁡(−x)

where n and m represent the feature mapping of the AG module input from the decoder and encoder pipelines, respectively. And pm, pn, pi, pk are the convolution kernels of size 1 × 1. bm,n, bint, bk represent the offset unit. ε1 and ε2 denote the RELU and sigmoid activation function respectively. Here ε2 limits the range between 0 and 1.

Figure 10 Diagram of attention gate (AG).

Finally, the attention coefficient σ is multiplied by the input feature map fi to present the output go as shown in Eq. (10): (10) go=σ×fi

From Fig. 11 and Table 5, results showed that the inclusion AG module improved the performance of the network (ADID-UNET), with segmentation accuracy of almost 97%. Therefore, by introducing the AG model, the network makes full use of the output feature information of encoder and decoder, which greatly reduces the probability of false prediction of small targets, and effectively improves the sensitivity and accuracy of the model.

Figure 11 The qualitative results of comparative experiments with or without Attention Gate in the network. Here (A–C) are three test images.

For each row, (D) denotes the ground truth, (E–I) illustrate the segmentation results from UNET, AG-UNET, DA-UNET, IDA-UNET and ADID-UNET, respectively. AG-UNET– the addition of AG module to UNET, DA-UNET–adding two dense networks and AG module to the network without including the IDC module. IDA-UNET refers to adding IDC and AG modules to the UNET without adding dense networks, and ADID-UNET indicates that dense networks, IDC and AG module are added to the network.

Table 5 The quantitative results of the comparison with or without the AG model experiment.

AG-UNET– the addition of AG module to UNET, DA-UNET–adding two dense networks and AG module to the network without including the IDC module. IDA-UNET refers to adding IDC and AG modules to the UNET without adding dense networks, and ADID-UNET indicates that dense network, IDC and AG module are added to the network.

Method	ACC	DC	Sen	Sp	Pc	AUC	F1	Sm	Eα	MAE	
UNET	0.9696	0.7998	0.8052	0.9957	0.8247	0.9347	0.8154	0.8400	0.9390	0.0088	
AG-UNET	0.9697	0.8020	0.8106	0.9962	0.8347	0.9571	0.8116	0.8511	0.9345	0.0087	
DA-UNET	0.9698	0.7754	0.7400	0.9959	0.8470	0.9274	0.7930	0.8334	0.9104	0.0091	
IDA-UNET	0.9698	0.7961	0.7834	0.9964	0.8469	0.9450	0.8126	0.8513	0.9437	0.0085	
ADID-UNET	0.9701	0.8031	0.7973	0.9966	0.8476	0.9551	0.8200	0.8509	0.9449	0.0082	

Experimence results

COVID-19 segmentation dataset collection and processing

Organizing a COVID-19 segmentation dataset is time-consuming and hence there are not many CT scan segmentation datasets. At present, there was only one standard dataset namely the COVID-19 segmentation dataset (MedSeg, 2020), which was composed of 100 axial CT scans from different COVID-19 patients. All CT scans were segmented by radiologists associated with the Italian Association of medicine and interventional radiology. Since the database was updated regularly, on April 13, 2020, another segmented CT scans dataset with segment labels from Radiopaedia was added. The whole datasets that contained both positive and negative slices (373 out of the total of 829 slices have been evaluated by a radiologist as positive and segmented), were selected for training and testing the proposed model.

The dataset consists of 1,838 images with annotated ground truth was randomly divided into 1,318 training samples, 320 validation samples, and 200 test samples. Since the number of training images is less, we expand the training dataset where we first merge the COVID-19 lung CT scans with the ground scene and then perform six affine transformations as mentioned in Krizhevsky, Sutskever & Hinton (2012). Later the transformed image is separated from the new background truth value and added to the training dataset as additional training images. Therefore, the 1,318 images of the training dataset are expanded, and 9,226 images are obtained for training. Figure 12 illustrates the data expansion process.

Figure 12 Data augmentation.

Illustration of vertical flipping process showing the expansion of the training dataset.

Segmentation evaluation index

The commonly used evaluation indicators for segmentation such as accuracy (ACC), precision (Pc), Dice coefficient (DC), the area under the curve (AUC), sensitivity (Sen), specificity (Sp) and F1 score (F1) were used to evaluate the performance of the model. These performance indicators are calculated as follows:

(1) For computing accuracy, precision, sensitivity, specificity, and F1 score we generate the confusion matrix where the definitions of true positive (TP), true negative (TN), false positive (FP), and false negative (FN) are shown in Table 6.

Table 6 Definition of TP, FP, FN, TN.

Category	Actual lesion	Actual non-lesion	
Predicted Lesion	True Position (TP)	False Position (FP)	
Predicted Non-Lesion	False Negative (FN)	True Negative (TN)	

(1) Accuracy (ACC): A ratio of the number of correctly predicted pixels to the total number of pixels in the image.

(11) Accuracy(ACC)=TP+TNTP+TN+FP+FN

(2) Precision (Pc): A ratio of the number of correctly predicted lesion pixels to the total number of predicted lesion pixels.

(12) Precision(Pc)=TPTP+FP

(3) Sensitivity (Sen): A ratio of the number of correctly predicted lesion pixels to the total number of actual lesion pixels.

(13) Sensitivity(Sen)=TPTP+FN

(4) F1 score (F1): A measure of balanced accuracy obtained from a combination of precision and sensitivity results.

(14) F1score(F1)=2×Pc×SenPc+Sen

(5) Specificity (Sp): A ratio of the number of correctly predicted non-lesion pixels to the total number of actual non-lesion pixels.

(15) Specificity(Sp)=TNTN+FP

(6) Dice coefficient (DC): Represents the similarity between the model segment output (Y) and the ground truth (X). The higher the similarity between the lesion and the ground truth, the larger the Dice coefficient and the better the segmentation effect. Dice coefficient is calculated as follows: (16) DiceCoefficient(DC)=2×(X∩Y)X+Y

Also, we use a Dice coefficient (Dice, 1945) loss (dice_loss) as the training loss of the model, the calculation is as follows: (17) Train Loss = Dice Coefficient Loss=1.0−2×(X∩Y)X+Y

(7) The area under the curve (AUC): AUC is the area under the receiver operating characteristic (ROC) curve. It represents the degree or the measure of separability and indicates the capability of the model in distinguishing the classes. Higher the AUC better is the segmentation output and hence the model.

In addition to the above widely used indicators, we also introduce the Structural metric (Sm) (Fan et al., 2017), Enhanced alignment metric (Eα) (Fan et al., 2018) and Mean Absolute Error (MAE) (Fan et al., 2020; Elharrouss et al., 2020) to measure the segmentation similarity with respect to the ground truth.

(8) Structural metric (Sm): Measures the structural similarity between the prediction map and ground truth segmented mask, it is more in line with the human visual system than Dice coefficient.

(18) Sm=(1−β)×Sos(Sop,Sgt)+β×Sor(Sop,Sgt)

where Sos stands for target perception similarity, Sor stands for regional perceptual similarity, β = 0.5 is a balance factor between Sos and Sor. And Sop stands for the final prediction result and Sgt represents the ground truth.

(9) Enhance alignment metric (Eα): Evaluates the local and global similarity between two binary maps computed based on Eq. (19): (19) Eα=1w×h∑iw∑jhα×(Sop(i,j),Sgt(i,j))

where w and h are the width and height of ground truth Sgt, (i,j) denotes the coordinates of each pixel in Sgt. α represents the enhanced alignment matrix: (20) α=2×(Sgt−Sgt)×(Sop−Sop)(Sgt−Sgt)2+(Sop−Sop)2

(10) Mean Absolute Error (MAE): Measures the pixel-wise difference between Sop and Sgt, defined as: (21) MAE=1w×h∑iw∑jh|Sop(i,j)−Sgt(i,j)|

Experimental Details

The ADID-UNET proposed in this paper is implemented in Keras framework and is trained and tested by using the workstation with NVIDIA GPU P5000. During the training process, we set the learning rate as lr=1×10−3, and Adam optimizer was selected as the optimization technique. The 9,226 training samples, 320 verification samples, and 200 test samples were resized to 128 × 128 and trained with a batch size of 32 for 300 epochs. Figures 13 and 14 shows the performance curves obtained for the proposed ADID-UNET during training, validation, and testing.

Figure 13 ADID-UNET training and validation performance index curve.

(A–D) indicate the loss of training and validation, accuracy, Dice coefficient, and sensitivity performance curves, respectively.

Figure 14 ADID-UNET training and validation performance index curve.

(A–D) indicate the training and validation of the specificity, F1 score, precision, and AUC performance curves, respectively.

Segmentation results and discussion

Qualitative results

To show the performance of the ADID-UNET model, we used 200 pairs of COVID-19 lung infection CT scans as test data, and the segmentation results are shown in Fig. 15. From the analysis of Fig. 15, it was found that the ADID-UNET model can accurately segment the COVID-19 lung infection areas from the CT scans, especially the smaller infected areas, and the segmentation result is very close to the ground truth. This illustrates the effectiveness of the proposed method for the segmentation of COVID-19 lung infection regions from CT scans. Moreover, we can also see that ADID-UNET can accurately segment the complicated infection areas (single COVID-19 lung infection areas and more complex uneven distribution infection areas) in CT scans, which further proves the power of the model proposed in this paper. In a word, the ADID-UNET model proposed in this paper can effectively and accurately segment COVID-19 lung infection areas with different sizes and uneven distribution, and the visual effect of segmentation is very close to the gold standard.

Figure 15 Visual comparison of the segmentation results of COVID-19 lung infection obtained from the proposed ADID-UNET.

Column 1 represents the original CT Image, column 2 the ground truth and column 3 the predicted segmentation results from the proposed network called the Attention Gate-Dense Network-Improved Dilation Convolution-UNET (ADID-UNET).

Further, we also compare the proposed model with other state-of-art segmentation models. From the results (Figs. A1 and A2 and Table 7), we can infer that the ADID-UNET model presents segmentation outputs closer to the ground truth. In contrast, the FCN8s network (Long, Shelhamer & Darrell, 2015) presents more under and over segmented regions. Further RAD-UNET (Zhuang et al., 2019a) presents comparable segmentation results but its effect is less pronounced for smaller segments. Analyzing the segmentation visual results from Figs. A1 and A2, we can clearly find that the ADID-UNET model proposed in this paper can accurately segment the COVID-19 lung infection regions than other state-of-the-art model with results close to the ground truth, which proves the efficacy of the proposed ADID-UNET model.

10.7717/peerj-cs.349/fig-A1 Figure A1 The visual comparison of the segmentation results of COVID-19 lung infection compared with other advanced models.

Here (A–C) illustrate three test images obtained from (MedSeg, 2020). For each row, (D) denotes the ground truth, and (E–K) illustrate the segmentation results from FCN8s (Long, Shelhamer & Darrell, 2015), UNET (Ronneberger, Fischer & Brox, 2015), Segnet (Badrinarayanan, Kendall & Cipolla, 2017), Squeeze UNET (Iandola et al., 2016), Residual UNET (Alom et al., 2018), RAD UNET (Zhuang et al., 2019a), ADID-UNET, respectively.

10.7717/peerj-cs.349/fig-A2 Figure A2 The visual comparison of the segmentation results of COVID-19 lung infection compared with other advanced models.

Here (A–C) illustrate three test images obtained from (MedSeg, 2020). For each row, (D) denotes the ground truth, and (E–K) illustrate the segmentation results from FCN8s (Long, Shelhamer & Darrell, 2015), UNET (Ronneberger, Fischer & Brox, 2015), Segnet (Badrinarayanan, Kendall & Cipolla, 2017), Squeeze UNET (Iandola et al., 2016), Residual UNET (Alom et al., 2018), RAD UNET (Zhuang et al., 2019a), ADID-UNET, respectively.

Table 7 Quantitative results of infected areas in the COVID-19 dataset. - - - means no relevant data in the original literature.

Method	ACC	DC	Sen	Sp	Pc	AUC	F1	Sm	Eα	MAE	
FCN8s	0.9666	0.6697	0.6692	0.9923	0.6860	0.9485	0.6724	0.7539	0.9134	0.0157	
UNET	0.9696	0.7998	0.8052	0.9957	0.8247	0.9347	0.8154	0.8400	0.9390	0.0088	
Segnet	0.9684	0.7408	0.7608	0.9937	0.7549	0.9492	0.7558	0.8080	0.9374	0.0125	
Squeeze UNET	0.9689	0.7681	0.7827	0.9946	0.7776	0.9446	0.7785	0.8227	0.9326	0.0107	
Residual UNET	0.9697	0.7924	0.7905	0.9961	0.8248	0.9444	0.8055	0.8397	0.9324	0.0094	
RAD UNET	0.9699	0.7895	0.7625	0.9970	0.8601	0.9419	0.8062	0.8475	0.9328	0.0096	
Fan et al. (2020)	- - -	0.7390	0.7250	0.9600	- - -	- - -	- - -	0.8000	0.8940	0.0640	
Elharrouss et al. (2020)	- - -	0.7860	0.7110	0.9930	0.8560	- - -	0.7940	- - -	- - -	0.0760	
Yan et al. (2020)	- - -	0.7260	0.7510	- - -	0.7260	- - -	- - -	- - -	- - -	- - -	
Zhou, Canu & Ruan (2020)	- - -	0.6910	0.8110	0.9720	- - -	- - -	- - -	- - -	- - -	- - -	
Chen, Yao & Zhang (2020)	0.8900	- - -	- - -	0.9930	0.9500	- - -	- - -	- - -	- - -	- - -	
ADID-UNET	0.9701	0.8031	0.7973	0.9966	0.8476	0.9551	0.8200	0.8509	0.9449	0.0082	

Quantitative results

Table 7, presents the performance scores for various indicators mentioned in “Experimence Results”. Here, for ADID-UNET the scores such as the Dice coefficient, precision, F1 score, specificity and AUC are 80.31%, 84.76%, 82.00%, 99.66% and 95.51%, respectively. Further, most of the performance indexes are above 0.8 with the highest segmentation accuracy of 97.01%. The above results clearly indicates that the proposed model presents segmentation outputs closer to ground truth annotations.

Discussion

The proposed model presents an improved version of the UNET model obtained by the inclusion of modules such as the dense network, IDC and the attention gates to the existing UNET (Ronneberger, Fischer & Brox, 2015) structure. The effectiveness of these additions were experimentally verified in “Methods”. Further, to summarize the effectiveness of the addition of each module to the UNET architecture, Table 8 tabulates the improvement at each stage of the addition. From Table 8, it is found that adding additional components to the UNET (Ronneberger, Fischer & Brox, 2015) structure can obviously improve the overall segmentation accuracy of the network. For example, with the inclusion of the dense networks (D-UNET), the metrics such as Dice coefficien (DC) and AUC reached 79.98% and 93.47%, respectively.

Table 8 The quantitative results showing percentages improvements of the model after adding additional components to UNET (Ronneberger, Fischer & Brox, 2015) structure.

D-UNET denotes dense networks with UNET structure, DID-UNET represents dense networks and improved dilation convolution to the structure of UNET, and ADID-UNET refers to proposed model with dense networks improved dilation convolution and attention gate modules to the UNET structure. ↑ indicates that the performance index is higher than that of UNET structure, ↓ indicates that the performance index is lower than that of UNET structure.

Method	ACC	DC	Sen	Sp	Pc	AUC	F1	Sm	Eα	MAE	
UNET	0.9696	0.7998	0.8052	0.9957	0.8247	0.9347	0.8154	0.8400	0.9390	0.0088	
D-UNET	0.9700	0.8011	0.8096	0.9966	0.8596	0.9492	0.8184	0.8528	0.9394	0.0083	
DID-UNET	0.9700	0.8023	0.7987	0.9964	0.8425	0.9549	0.8241	0.8447	0.9374	0.0084	
ADID-UNET	0.9701	0.8031	0.7973	0.9966	0.8476	0.9551	0.8200	0.8509	0.9449	0.0082	
Improvement of D-UNET	↑0.04%	↑0.13%	↑0.44%	↑0.09%	↑3.49%	↑1.45%	↑0.30%	↑1.28%	↑0.04%	↓0.05%	
Improvement of DID-UNET	↑0.04%	↑0.25%	↓0.65%	↑0.07%	↑1.78%	↑2.02%	↑0.87%	↑0.47%	↓0.16%	↓0.04%	
Improvement of ADID-UNET	↑0.05%	↑0.33%	↓0.79%	↑0.09%	↑2.29%	↑2.04%	↑0.46%	↑1.09%	↑0.59%	↓0.06%	

Further, the inclusion of the IDC improved the scores further (DID-UNET). Finally, the proposed model with dense network, IDC and the AG modules (namely ADID-UNET) presented the best performance scores and provided an improvement of 0.05%, 0.33%, 2.29%, 2.04% and 1.09% for metrics such as accuracy, DC, precision, AUC and structural metric respectively when compared to traditional UNET architecture.

Furthermore, from Figs. A1 and A2, it is obvious that ADID-UNET performs better than other well-known segmentation models in terms of visualization. Specifically, ADID-UNET can segment relatively smaller infected regions which is of great significance for clinical accurate diagnosis of COVID-19 infection location. The use of (a) dense networks instead of traditional convolution and max-pooling function, (b) inclusion of improved dilation convolution module between the encoder-decoder pipeline and (c) the presence of attention gate network in the skip connections have presented accurate segmentation outputs for various types of COVID-19 infections (GGO and pulmonary consolidation). However, ADID-UNET still has room for improvement in terms of Dice coefficient and sensitivity and also computational costs which can be researched in future.

Conclusion

The paper proposes a new variant of UNET (Ronneberger, Fischer & Brox, 2015) architecture to accurately segment the COVID-19 lung infections in CT scans. The model, ADID-UNET includes dense networks, improved dilation convolution, and attention gate, which has strong feature extraction and segment capabilities. The experimental results show that ADID-UNET is effective in segmenting small infection regions, with performance metrics such as accuracy, precision and F1 score of 97.01%, 84.76%, and 82.00%, respectively. The segmentation results of the ADID-UNET network can aid the clinicians in faster screening, quantification of the lesion areas and provide an overall improvement in the diagnosis of COVID-19 lung infection.

Appendix

We describe the abbreviations of this paper in detail, as shown in Table A1.

Table A1 An explanation of the acronyms that appears in this article.

Abbreviation	Explanation	
D-UNET	Inclusion of Dense networks to the UNET structure	
AG-UNET	Inclusion of Attention gate module to the UNET structure	
DA-UNET	Inclusion of both dense networks and attention gate module to the UNET structure	
IDA-UNET	Inclusion of Improved dilation convolution and Attention Gate module to the UNET structure	
DID-UNET	Inclusion of dense networks and improved dilation convolution to the UNET structure	
ADID-UNET	Inclusion of dense networks, Improved dilation convolution and Attention Gate modules to the UNET structure	

Supplemental Information

Supplemental Information 1 Code, test data, and results.

Click here for additional data file.

Supplemental Information 2 ADID-Model Usage and Results Verification.

Click here for additional data file.

We are very grateful to the Italian Society of medicine and interventional radiology, Radiopedia, and Ma et al. (2020) for providing the COVID-19 CT scan segmentation database.

Additional Information and Declarations

Competing Interests

Author Contributions

Data Availability

The authors declare that they have no competing interests.

Alex Noel Joseph Raj conceived and designed the experiments, performed the experiments, analyzed the data, performed the computation work, prepared figures and/or tables, authored or reviewed drafts of the paper, and approved the final draft.

Haipeng Zhu conceived and designed the experiments, performed the experiments, analyzed the data, performed the computation work, prepared figures and/or tables, authored or reviewed drafts of the paper, and approved the final draft.

Asiya Khan analyzed the data, prepared figures and/or tables, authored or reviewed drafts of the paper, and approved the final draft.

Zhemin Zhuang conceived and designed the experiments, analyzed the data, prepared figures and/or tables, and approved the final draft.

Zengbiao Yang conceived and designed the experiments, performed the experiments, analyzed the data, performed the computation work, prepared figures and/or tables, and approved the final draft.

Vijayalakshmi G. V. Mahesh performed the experiments, analyzed the data, prepared figures and/or tables, authored or reviewed drafts of the paper, and approved the final draft.

Ganesan Karthik analyzed the data, performed the computation work, prepared figures and/or tables, authored or reviewed drafts of the paper, and approved the final draft.

The following information was supplied regarding data availability:

Code and data is available in the Supplemental Files.

The trained model and usage procedure are available at GitHub:

https://github.com/jalexnoel/ADID-UNET.git.

The dataset is available at MedSeg through Figshare:

MedSeg; Jenssen, Håvard Bjørke; Sakinis, Tomas (2021): MedSeg Covid Dataset 1. figshare. Dataset. DOI 10.6084/m9.figshare.13521488.v2.

MedSeg; Jenssen, Håvard Bjørke; Sakinis, Tomas (2021): MedSeg Covid Dataset 2. figshare. Dataset. DOI 10.6084/m9.figshare.13521509.v2.

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
