# Peer review of "ADID-UNET—a segmentation model for COVID-19 infection from lung CT scans"

_PeerJ Computer Science, doi:10.7717/peerj-cs.349_

## Round 0.1 · original submission · Major Revisions

Consider all the reviewers' comments and revise the manuscript accordingly.

Reviewer 1 ·

Basic reporting

This manuscript proposes a novel methodology for segmenting the covid-19 infections This manuscript is written well.
Read the paper carefully to remove the typos and grammatical errors
Some sentences are not understandable to the reviewers Example :Summary of CT scan segmentation methods in COVID-19 application based on deep learning in this paper.
This line has grammatical error
The authors has to explain what are the features extracted for the purpose of segmentation?
Figure 15.Visual comparison of the segmentation results of COVID-19 lung infection should be explained
Check the performance measure as it confuses the reviewers. Make it more clear in the abstract

Experimental design

Experimental results were satisfactory. But in the abstract , the performance measure is mentioned as 80% where ares in the MS, it is different. Check this throughout the MS
Results were clearly explained
Reviewers were in dilemma to validate the accuracy since the proposed methodology has many metrics used to validate the accuracy..
The supplementary files should be executable so that the reviewers can validate the results since this works belongs to Covid-19. Python files were not sufficient.
Submit the coding as Matlab file so that the reviewers can validate the proposed methodology and its accuracy

Validity of the findings

Findings were good. But different types of results were confusing the reviewers.
Arrange all the results in its appropriate sections.
A thorough experiment is done
Reviewers has to validate the proposed methodology via the Coding
All the data were provided. Provide the Matlab coding as supplementary file

Additional comments

This manuscript proposes a novel methodology for segmenting the covid-19 infections This manuscript is written well.
Read the paper carefully to remove the typos and grammatical errors
Some sentences are not understandable to the reviewers Example :Summary of CT scan segmentation methods in COVID-19 application based on deep learning in this paper.
This line has grammatical error
The authors has to explain what are the features extracted for the purpose of segmentation?
Figure 15.Visual comparison of the segmentation results of COVID-19 lung infection should be explained
the performance measure is mentioned as 80% where ares in the MS, it is different. Check this throughout the MS
Check the performance measure as it confuses the reviewers. Make it more clear in the abstract
Submit the coding as Matlab file so that the reviewers can validate the proposed methodology and its accuracy.
Arrange the results accordingly.

Reviewer 2 ·

Basic reporting

The topic and manuscript appears interesting theoretically and in terms of results, however the paper requires improvements in terms of the following comments:
1) The main manuscript should be checked for typos
2) The results, should be better detailed in one section so that it can be easily refereed as well as understandable to the readers.
3) There were more results and it should be validated by the reviewers.
4) The executable program of the proposed methodology should must be presented as supplementary file. Change the current file format.
5) The validation would be better supported if you would add the executable programs and also with some image datasets.
6) Transitions from section to section should be smoother. Intermediate results can be changed to the results and discussion section.

Experimental design

Experiments were performed well. Research question is well defined.
Methods were described well

Validity of the findings

The findings were novel
All the data related to the validity is provided
Conclusions are well stated

Additional comments

1) The main manuscript should be checked for typos
2) The results, should be better detailed in one section so that it can be easily refereed as well as understandable to the readers.
3) The executable program of the proposed methodology should must be presented as supplementary file. Change the current file format.
5) The validation would be better supported if you would add the executable programs and also with some image datasets.
6) Transitions from section to section should be smoother. Intermediate results can be changed to the results and discussion section.

---

## Round 0.2 · accepted · Accept

This manuscript has been Accepted for publication. Congratulations!

Reviewer 1 ·

Basic reporting

All the revisions were done

Experimental design

All the revisions were done

Validity of the findings

All the revisions were done

Additional comments

I can find that all my comments were updated in the revised manuscript

Reviewer 2 ·

Basic reporting

The manuscript is revised based on my previous comments

Experimental design

The manuscript is revised based on my previous comments

Validity of the findings

The manuscript is revised based on my previous comments

Additional comments

The manuscript is revised based on my previous comments. All the revisions were done.